# Research on Sequential Decision-Making of Major Accidents with Incomplete Information

**Dengyou Xia [1], Changlin Chen [1,*], Ce Zheng [2], Jing Xin [1] and Yi Zhu [1]**

1   Hebei Key Laboratory of Emergency Rescue Technology, China People's Police University, Langfang 065000, China; xiadengyou@cppu.edu.cn (D.X.); xinjing@cppu.edu.cn (J.X.); zhuyi01@cppu.edu.cn (Y.Z.)
2   Heihe Fire and Rescue Detachment, Heihe 164300, China; 17545585119@163.com
*   Correspondence: chenchanglin@cppu.edu.cn

**Abstract:** In order to solve the problem of emergency decision-making with incomplete information and deal with the accident information in different time series at the scenes of major accidents, this paper proposes a method of sequential decision-making by utilizing the relevant knowledge of D-S evidence theory and game theory. Firstly, we took an oil tank fire accident as an example and sorted out historical cases and expert experiences to establish a logical relationship between key accident scenes and accident scene symptoms in the accident. Meanwhile, we applied the logistic regression analysis method to obtain the basic probability distribution of each key accident scene in the oil tank fire, and on this basis, we constructed an evidence set of the fire. Secondly, based on the D-S evidence theory, we effectively quantified the knowledge uncertainty and evidence uncertainty, with the incomplete and insufficient information taken as an evidence system of the development of key accident scenes to construct a situation prediction model of these accident scenes. Thirdly, based on the game theory, we viewed emergency decision-makers and major accidents as two sides of the game to compare and analyze accident states at different time points and solve the contradiction between loss costs of decision-making and information collection costs. Therefore, this paper has provided a solution for the optimization of accident schemes at different time stages, thus realizing the sequential decision-making at the scenes of major accidents. Furthermore, we combined the situation prediction model with sequential decision-making, with the basic steps described below: (1) We drew up an initial action plan in the case of an extreme lack of information; then, we (2) started to address the accident and constructed a framework of accident identification, (3) collected and dealt with the continuously added evidence information with the evolution of the accident, (4) calculated the confidence levels of key accident scenarios after evaluating different evidence and then predicted the accident state in the next stage, and (5) calculated the profit–loss ratio between the current decision-making scheme and the decision-making scheme of the next stage. Finally, we (6) repeated steps (3) to (5) until the accident completely vanished. We verified the feasibility of the proposed method with the explosion accident of the Zhangzhou P.X. project in Fujian on 6 April used as an example. Based on the D-S evidence theory, this method employs approximate reasoning on the incomplete and insufficient information obtained at the scenes of major accidents, thus realizing the situation prediction of key scenes of these accidents. Additionally, this method uses the game theory to solve the contradiction between decision-making loss costs and information collection costs, thus optimizing the decision-making schemes at different time stages of major accidents.

**Keywords:** sequential decision-making; incomplete information; approximate reasoning; situation prediction; D-S evidence theory

## 1. Introduction

In recent years, the rapid industrial development in China has brought about great convenience to people's lives. However, with the gradual progress of industrialization and

urbanization, safety accidents have occurred frequently in various industries due to natural reasons or human errors. Additionally, safety accidents have posed a great threat to the national economic development and people lives, hindering the social development to a certain extent. For example, on 21 March 2019, the explosion accident of Xiangshui Chemical Company, Yancheng, China, killed 78 people and seriously injured 76, with 640 people hospitalized and a direct economic loss of 1.986 billion yuan. On 10 January 2021, ten people were killed, and one person was reported missing in a major explosion accident at Hushan Gold Mine in Qixia City, China, with a direct economic loss of 68.4733 million yuan. On 18 April 2023, a sudden fire broke out in Changfeng Hospital, Fengtai District, Beijing, China, killing 29 people and resulting in an extremely bad social impact. Nowadays, for the national emergency departments and academic circles, how to make effective emergency decisions in case of major accidents, such as fires and explosions, and minimize accident losses has always been an important problem that needs to be urgently addressed. Therefore, scholars at home and abroad have conducted a lot of research on the emergency decision-making problem of major accidents. In 2007, Fan [1] proposed the concept of emergency decision-making and pointed out that emergency decision-making is a dynamic process in which emergency commanders collect and sort out relevant information in the first place and formulate emergency plans that are immediately implemented and timely adjusted. Based on the systematic framework of emergency management of information-based railway, Liu [2] introduced a regret theory, in which the relevance of decision attributes and the rational characteristics of decision-makers have been considered, and a multi-attribute group decision-making method involving regret avoidance has been put forward. Liu et al. [3] proposed a fuzzy multi-attribute and synergy-based emergency decision-making method for addressing network public opinion emergencies. To address the problems of uncertain information and bounded rationality of decision-makers during the emergency decision-making process, Li et al. [4] proposed a decision-making method of emergency responses based on the prospect theory and probabilistic language terms. To solve the problem of incomplete preference information of experts during their decision-making processes, Xu et al. [5] studied the influences of public risk perception on the quality of emergency responses at different stages. On the basis of the conventional group decision-making process, Wu [6] combined the D-S evidence theory with factors such as expert preference to study and analyze the group decision-making performances of earthquake emergency rescues. From the perspective of prediction and group analysis, those studies described above have primarily applied theoretical methods, such as optimization models, to perform various research on the decision-making in emergency accidents, solving the problem of fuzzy and random decision-making of emergency responses to a large extent.

After the occurrence of a major accident, its state will continuously change with time. Therefore, emergency decision-making in a major accident is a dynamic process, with the results of a previous decision directly influencing a later one. Therefore, dynamic decision-making, also called sequential decision-making, has become a hot research topic at home and abroad. For example, by combining the Bayesian method with the dynamic game theory, Ding et al. [7] proposed a generation method for urban rainstorm emergency plans and illustrated the method with an algorithmic example. Considering that the psychological activities of decision-makers will constantly change under the influences of decision-making factors and effects, Xu et al. [8,9] put forward a dynamic decision-making method based on big data. Integrating fuzzy decision-making ideas with the sequential game method, Song [10] constructed an emergency decision-making approach to smart city disasters. Using a multi-dimensional scenario space method, Wang et al. [11] systematically analyzed the scenario evolution mechanisms of environmental emergency accidents and constructed an emergency decision-making model of environmental emergency accidents based on case reasoning. By introducing the optimal decision transition into the Logic Petri Net, Li et al. [12] constructed a game decision Petri net, which can better describe such systematic variables as uncertain elements and utility functions, thus optimizing the selection of a contingency plan. With the introduction of Bayesian theory, Liu [13]

constructed a sequential game model of off-site emergency decision-making for nuclear accidents, satisfying the requirements of generating disposal plans during the grouping phase of emergency decision-making in nuclear accidents. Most existing studies on dynamic decision-making have focused on the influences of multi-target and multi-subject coordination in emergency responses of dynamic decision-making, with a lack of research on the selection and adjustment of decision-making during the sub-stage of the disposal scheme [14,15]. Studies on accident evolution reasoning are isolated from studies on emergency decision-making, and most scholars have even completely separated these two topics in their studies. Meanwhile, few studies have considered the condition of incomplete information in major accident responses, with a lack of an effective decision-making method for major accidents with incomplete information. Therefore, in this paper, we have taken the emergency decision-making of major accidents as the research object and applied the situation prediction model based on the D-S evidence theory (Dempster–Shafer evidence theory) to correct the subjective deviation of decision-makers with incomplete information. Meanwhile, we used the idea of sequential game theory to compare the profit–loss ratios of decision-making schemes at different practical stages, thus realizing the emergency decision-making of major accidents with incomplete information.

## 2. Evidence Set Construction of Major Accidents

### 2.1. Definition of Key Accident Scenarios and Symptoms

Key accident scenarios refer to the on-site events or factors that have a decisive influence on the transformation, spread, derivation, coupling, and mutation mechanisms of major accidents. These scenarios promote the state and stage transformation of major accidents, interact with the composition of on-site emergency rescue forces, or play a leading role in the objectives of emergency decision-making activities. Key accident scenarios are a key factor that decision-makers need to focus on when setting decision-making objectives and making decision-making plans at the scenes of major accidents, and an important part and manifestation of major accidents themselves. Key accident scenario symptoms refer to the specific events or states leading to the occurrences of key accident scenarios. There is an internal relationship between accident scenario symptoms and accident scenarios. Objectively, it is a certain logical relationship that can be easily discovered through the collection of on-site information. Therefore, it can provide an important basis for approximate reasoning of major accident scenes through enriched field information collection means, sufficient key accident scenario symptoms acquired, and logical relationships between key accident scenarios and key accident scenario symptoms explored.

### 2.2. Construction Methods of Major Accident Evidence Sets

In order to obtain the evidence sets of major accident scenarios, it is necessary to construct the accident scenarios. The construction of accident scenarios refers to the processes of extracting different scenario elements and constructing corresponding scenarios through data induction, integration, statistics, and analysis. The technical route of major accident scenario construction primarily contains the following three steps: The first step is to collect and decompose data. In this step, a large number of major accident cases are collected and analyzed, and these accidents are decomposed into many accident scenarios with their basic data obtained. The second step is to evaluate and converge the data. In this step, mining, cleaning, clustering, and absorbing of the massive amounts of data obtained are performed by centering on major accidents and relying on professionals and methods. The third step is to integrate and describe the data. In this step, through data collection and appraisal of major accident scenarios, several most common accident scenarios are singled out. That is, the scenario space of accidents. According to the approximate reasoning method, key accident scenarios of major accidents can be identified through the study of historical materials or interviews with experts in related fields. Then, key accident scenario symptoms corresponding to each key accident scenario can be identified. Key accident scenario symptoms refer to the direct accident information available to decision-makers

at the scenes of major accidents. This accident information can reflect the characteristics of accident evolution. With all possible key accident scenarios at the scenes of major accidents and their corresponding key symptoms summarized, their logical relationships can be constructed.

*2.3. Construction of Accident Scenario Evidence Sets—Taking an Oil Tank Fire as an Example*

With an oil tank fire taken as an example, through analysis of historical cases and interviews with experts in related fields, this study has identified key accident scenarios of major accidents, as well as scenario symptoms corresponding to each key accident scenario, with logical relationships between major accident symptoms and key accident scenarios constructed.

Figure 1 shows logical relationships between accident symptoms and key accident scenarios of the oil tank fire accident.

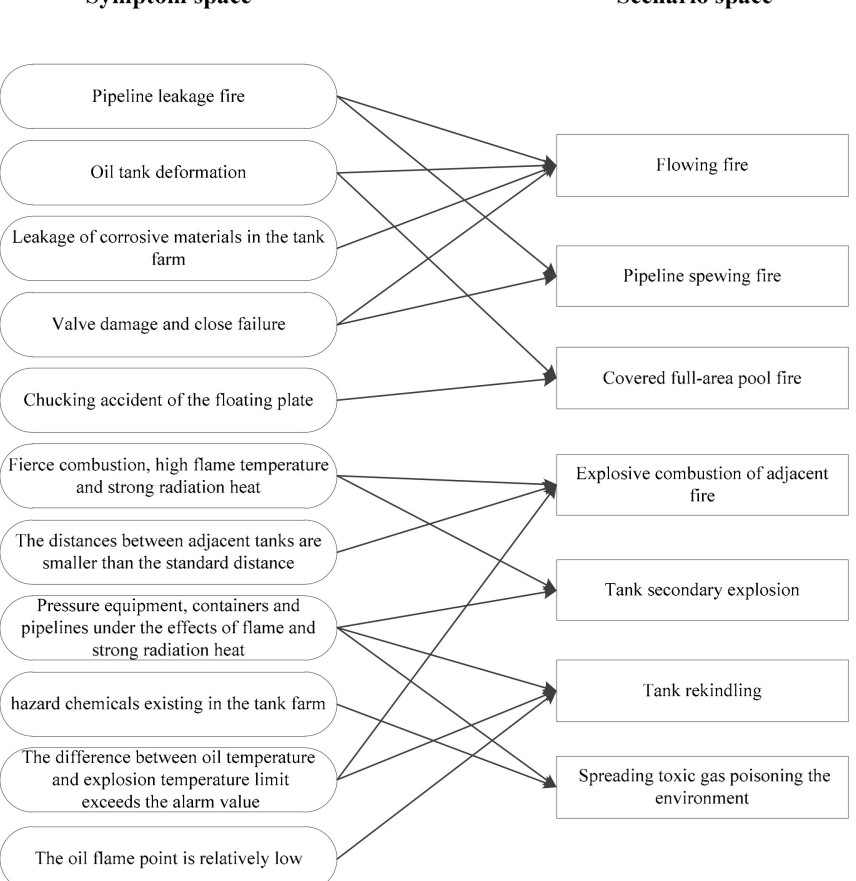

**Figure 1.** Logical relationships between accident symptoms and key accident scenarios.

On the basis of the identified logical relationships between accident symptoms and key accident scenarios, a scenario space was extracted as the dependent variable, and the accident symptom corresponding to that scenario was taken as the independent variable. Then, logistic regression analysis with binary variables [16] was performed in this study. Table 1 shows the results of the logistic regression analysis using SPSS statistical software (https://www.ibm.com/products/spss-statistics, accessed on 15 December 2023), with flowing fire as the dependent variable, and pipeline leakage fire, oil tank deformation, leakage of corrosive substances in the tank farm, and valve damage and close failure as independent variables.

**Table 1.** Logistic regression analysis results, with flowing fire as the dependent variable.

| Symptom Space | B | S.E. | Wals | df | Sig | Exp (B) | EXP (B) 95% C.I. | |
|---|---|---|---|---|---|---|---|---|
| | | | | | | | Lower Limit | Upper Limit |
| Pipeline leakage fire | 0.917 | 22,670.964 | 0 | 1 | 0.290 | 2.358 | 0 | |
| Oil tank deformation | 1.414 | 0.826 | 2.928 | 1 | 0.087 | 4.114 | 0.814 | 20.783 |
| Leakage of corrosive materials in the tank farm | 1.088 | 0.915 | 1.413 | 1 | 0.034 | 2.967 | 0.494 | 17.829 |
| Valve damage and close failure | 0.006 | 0.911 | 0 | 1 | 0.045 | 1.006 | 0.169 | 5.999 |
| Constant | 19.157 | 22,670.964 | 0 | 1 | 0.999 | 208,919,956.9 | | |

In Table 1, *B* represents the regression coefficient, *S.E.* represents the standard error, *Wals* represents the chi-square value, *df* represents the degrees of freedom, *Sig* represents the significance value, and Exp (*B*) represents the odds ratio.

From Table 1, it can be seen that for the key accident scenario of oil tank deformation causing a flowing fire, the calculated value of Exp (*B*) is 4.114, indicating that the probability of oil tank deformation causing a flowing fire is 4.114 times that of an oil tank with no deformation causing a flowing fire. Therefore, it can be concluded that the probability of oil tank deformation causing a flowing fire has a value of 0.804. Similarly, it can be calculated that the probability of corrosive substance leakage in the tank farm causing a flowing fire has a value of 0.748, the probability of pipeline leakage fire causing a flowing fire has a value of 0.702, and the probability of valve damage and close failure causing a flowing fire has a value of 0.501. Through the above analysis, the probability of accident symptoms corresponding to each key accident scenario was calculated and summarized, with the completed oil tank fire evidence set listed in Table 2.

**Table 2.** Evidence set of the oil tank fire.

| Accident Scenario Space | Scenario Symptom Space | Sig Value | Exp (B) Value | Basic Probability |
|---|---|---|---|---|
| Flowing fire ($S_1$) | Pipeline leakage fire | 0.29 | 2.358 | 0.702 |
| | Oil tank deformation | 0.087 | 4.114 | 0.815 |
| | Leakage of corrosive materials in the tank farm | 0.034 | 2.967 | 0.748 |
| | Valve damage and close failure | 0.045 | 1.006 | 0.501 |
| Pipeline spewing fire ($S_2$) | Pipeline leakage fire | 0.047 | 1.875 | 0.652 |
| | Valve damage and close failure | 0.054 | 2.385 | 0.705 |
| Hidden dead corner fire ($S_3$) | Oil tank deformation | 0.025 | 8.222 | 0.892 |
| | Chucking accident of the floating plate | 0.033 | 4.760 | 0.826 |
| Explosive combustion of adjacent tank ($S_4$) | Fierce combustion, high flame temperature, and strong radiant heat | 0.407 | 2.131 | 0.681 |
| | Difference between oil temperature and explosion temperature limit exceeds the alarm value | 0.069 | 6.292 | 0.863 |
| | Distances between adjacent tanks are smaller than the standard distance | 0.079 | 3.171 | 0.760 |

**Table 2.** *Cont.*

| Accident Scenario Space | Scenario Symptom Space | *Sig* Value | Exp (*B*) Value | Basic Probability |
|---|---|---|---|---|
| Tank secondary explosion ($S_5$) | Fierce combustion, high flame temperature, and strong radiant heat | 0.041 | 3.867 | 0.795 |
| | Pressure equipment under the effects of flame and strong radiation heat | 0.046 | 2.529 | 0.717 |
| | Not favorable wind direction for the on-site firefighting and rescue activity | 0.026 | 4.909 | 0.831 |
| Tank rekindling ($S_6$) | Difference between oil temperature and explosion temperature limit exceeds the alarm value | 0.195 | 3.415 | 0.773 |
| | The oil flame point is relatively low | 0.039 | 1.529 | 0.605 |
| | Not favorable wind direction for the on-site firefighting and rescue activity | 0.042 | 1.199 | 0.528 |
| Spreading toxic gas, poisoning the environment ($S_7$) | Hazardous chemicals existing in the tank farm | 0.055 | 2.830 | 0.739 |
| | Not favorable wind direction for the on-site firefighting and rescue activity | 0.058 | 3.427 | 0.774 |

## 3. Situation Predictions of Major Accidents

### 3.1. D-S Evidence Theory

D-S evidence theory [17] is a kind of uncertainty processing method with the core content of uncertainty measurement that can effectively reflect uncertain events. Through the reasoning and fusion of uncertainty, in this theory, uncertainty can be reduced, and decision accuracy can be improved. No prior knowledge is required in this method and the uncertainty problem can be resolved with interval estimation. Therefore, it can effectively solve the problem of incomplete and uncertain information in major accident decision-making.

### 3.2. Situation Prediction Method Based on D-S Evidence Theory

The occurrence and development of a major accident is a systematic process consisting of information from different periods. Information on the accident scenes during different periods is often incomplete and insufficient. However, this information can reflect scenario states of the accident at different stages from different aspects. That is, the accident scenario symptoms. Through these symptoms, evidence can be discovered to judge accident states. Among the evidence, the preliminary judgment made by the decision-maker on the accident scene can be viewed as an evidence system, and some historical data, data obtained with various technical means, expert knowledge, and empirical data can be included in the evidence system, too. The D-S evidence combination rules can be used to fuse the accident evidence and, therefore, the prediction results of accident situations can be obtained. With more data information obtained and used as the basis for the next step in situation prediction, the development trends of new accidents can be predicted. Therefore, foundations can be laid for on-site commanders to plan in advance and make deployment decisions so as to implement scientific and effective emergency decision-making. Figure 2 shows the model framework.

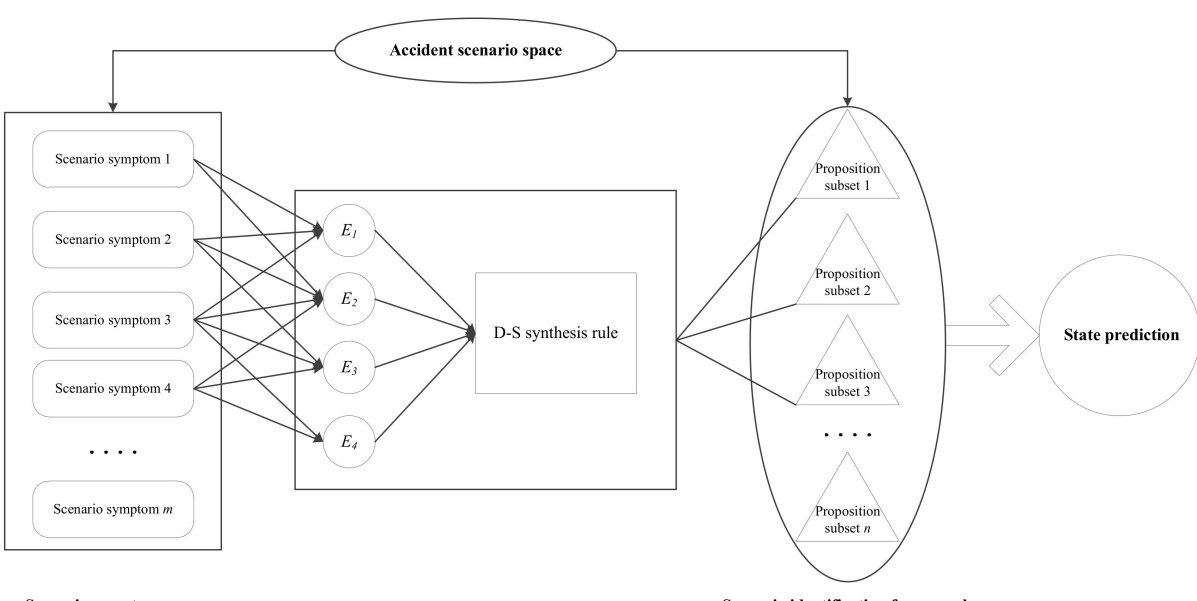

**Figure 2.** Model framework of situation prediction based on the D-S evidence theory.

(1) Analysis of scenario and symptom space: Based on on-site conditions, possible scenarios on the scenes of accidents are analyzed, primary disposal targets are identified, and accident identification frameworks are constructed.

(2) Processing of evidence information: The basic probability distribution shown in the evidence set is based on the analysis and summary of a large number of accident cases and is obtained through a certain method. Its significance lies in the basic grasp of relevant key accident scenarios. The scene of a major accident is complex and dynamic, and the prediction and emergency decision-making in the field situation of a major accident is itself an approximate reasoning with a probability sum not equal to 1. Therefore, when receiving information on accident symptoms, on-site decision-makers need to assign a value of confidence level to the symptom information itself. For example, when a fire breaks out in a storage tank in a chemical industry park, the accident symptom information "pipeline leakage fire" is obtained through on-site reconnaissance, but the decision-maker may not be able to completely determine the real combustion state of the pipeline. Under such a circumstance, the confidence level of the decision-maker on the on-site situation can be defined as $Pz$. Meanwhile, when receiving the information of the key scenario symptom of "pipeline leakage fire", the decision-maker can match the information with key accident scenarios provided in the evidence set. Under the assumption that these matched accident scenarios are flowing fire ($S_1$) and pipeline spewing fire ($S_2$), the basic probabilities of these two key accident scenarios, which are defined as $P(S_1)$ and $P(S_2)$, can be obtained. However, due to uncertain on-site situations, it is necessary to assign values to the probabilities of no occurrences of these two accident scenarios, which are $P'(S_1)$ and $P'(S_2)$. Thus, a piece of evidence information has been processed.

(3) Obtaining new accident evidence: Step (2) is repeated to process the newly obtained evidence.

(4) Evidence fusion: On the basis of combination rules of evidence theory, the synthesized reliability function and plausibility function (*F.B.* ($E_m$) and *F.P.* ($E_m$)) of each key accident scenario with different evidence can be obtained.

For an identification framework *A*, the synthesis rule of two or more mass functions can be described as follows:

$$(m_1 \oplus m_2 \oplus \cdots \oplus m_n)(A) = \frac{1}{1-K} \sum_{A_1 \cap \cdots \cap \cdots \cap A_n} m_1(A_1) m_2(A_2) \cdots m_n(A_n) \qquad (1)$$

where:

$$K = \sum_{A_1 \cap \cdots \cap A_n} m_1(A_1) m_2(A_2) \cdots m_n(A_n) \tag{2}$$

Its reliability, $F_B (A)$, is defined as the sum of all basic probabilities corresponding to all subsets in the framework $A$. That is,

$$\begin{aligned} &F_B : 2^\Omega \to [0,1] \\ &F_B(A) = \sum_{B \leq A} M(B), A \subseteq \Omega \end{aligned} \tag{3}$$

The reliability function, $F_B$ (bel function), also called the lower bound function, reflects all the confidence levels of $A$. The plausibility function, $F_P$, also called the upper bound function, reflects all the non-false confidence levels of $A$. That is, the uncertainty measurement of $A$. The reliability and plausibility functions present the following relationship:

$$F_P(A) \geq F_B(A), A \subseteq \Omega \tag{4}$$

The uncertainty of $A$ can be expressed with the following formula:

$$\mu(A) = F_P(A) - F_B(A) \tag{5}$$

(5) Situation prediction and emergency decision-making: A confidence interval is used to compare the threshold values set by the decision-maker. Generally, threshold values are determined by experts in the field based on real situations of the scenes to realize the situation prediction of key accident scenarios.

## 4. Sequential Decision-Making in Major Accidents

Field situations of major accidents change with time. Therefore, decision-makers should adjust their decisions upon the evolution of accidents. This indicates that on-site decision-making of major accidents belongs to the typical type of sequential decision-making [18]. Based on the situation prediction model constructed with the D-S evidence theory, decision-makers can integrate all kinds of information obtained from the scenes of major accidents and predict and analyze the development trends of accidents. Therefore, they can predict possible key accident scenarios and symptoms, thus formulating their proper disposal plans. However, during on-site decision-making processes of major accidents, it is impossible for decision-makers to determine the types of accidents immediately. They can only speculate on the accident types based on the on-site situations and take action accordingly. At the same time, the evolution patterns of major accidents will sustain before decision-makers act. Therefore, major accidents will continue to evolve under the effects of measures taken by decision-makers. Under such a circumstance, major accidents can be approximately viewed as a participating party with a game "ability". Accordingly, game theory was introduced in this paper to analyze the sequential decision-making processes of major accidents.

### 4.1. Sequential Decision-Making

Sequential decision-making is a type of decision-making with uncertain relevant parameters varying with time. During the process of sequential decision-making, a decision-maker should consider as many problems that would occur as possible [19,20] and formulate corresponding countermeasures in a timely manner. The development of major accident scenarios follows the evolution patterns of accidents themselves and, at the same time, is determined by the disposal plans made by decision-makers. With the development of accident situations, decision-makers should comprehensively consider various factors and make decisions at different stages of time series. On-site decision-making in major accidents bears the following characteristics:

(1) The systems being studied are dynamic. That is, the states of these systems vary with time and can be observed periodically (or continuously).

(2) Decision-making schemes are generated within certain time series. Due to insufficient information obtained, the implementation scheme adopted at an early stage of an accident will be quite unreliable. However, if more time is spent on on-site data collection, the scale of the accident could be expanded, and the optimal accident disposal time could be missed.

(3) The possible system state in the next step (or in the future) is random or uncertain. Subsequent situations that could occur in accident scenes can only be predicted scientifically on the basis of the on-site information collected, and in such a case, the state probability of each key accident scenario can be calculated.

### 4.2. Generation Method of Sequential Decision-Making Schemes

The conventional selection method of decision-making schemes can be described as follows: For a specific accident scenario, there could be different alternative schemes at the same time. By comparing the possible consequences of these alternative schemes, a decision-maker can select the scheme that causes a minimum loss at that moment or is more likely to make the accident evolve toward the most favorable direction. Such a process is called scheme optimization. However, on-site information available at the initial stages of major accidents is extremely scarce, and only with the progress of time can more information be obtained. Therefore, under such a circumstance, there will be a contradiction, as shown in Figure 3.

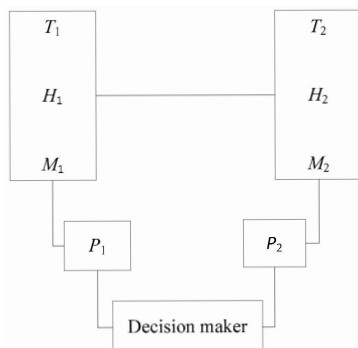

**Figure 3.** Sequential decision-making paradox.

It can be seen that at the $T_1$ moment, under a specific scenario of the accident, the information available to the decision-maker is $H_1$ and the loss caused at this moment is $M_1$. In such a case, after comprehensively analyzing field situations, the decision-maker adopts an optimal scheme of $P_1$. At the $T_2$ moment, the accident evolves into another specific scenario, the information available to the decision maker is $H_2$, and the loss caused at this moment is $M_2$. After comprehensively analyzing field situations, the decision-maker adopts an optimal scheme of $P_2$ at this moment. Between these two moments, the amount of information available at moment $T_1$ is less, with a smaller loss caused at that moment. On the contrary, more information is available at moment $T_2$. However, more losses are caused at moment $T_2$, compared to the losses caused at moment $T_1$. Therefore, it is necessary for the decision-maker to decide at that moment whether he or she should spend more time to collect information for developing a more scientific disposal scheme. In order to solve the contradiction mentioned above, it is necessary to introduce a decision-making loss function in the selection of an optimal solution scheme. Assume that at a certain moment, there are J key accident scenarios in a major accident, the *j*th scenario is denoted as $S_j$ ($j$ = 1, 2, ..., J), and the corresponding optimal solution scheme is $P_i$ ($i$ = 1, 2, ..., J). The inverse number of the decision loss cost ($B_j$) corresponding to the solution scheme adopted by the decision-maker is used to represent the utility function ($U_{ji}$) of the scheme. That is,

$$U_{ji} = -\begin{cases} B_{ji} = 0, j = i \\ B_{ji} \neq 0, j \neq i \end{cases} \tag{6}$$

The cost of information collection at each stage is recorded as $C_k$, and the purpose of collecting accident information is to predict the probabilities of various possible states of accidents. That is, to test whether the decision-maker has adopted a corresponding optimal emergency solution scheme under a true accident state at a specific stage. The decision loss cost of sequential decision-making is defined as the difference between the benefit of the selected scheme and the benefit of the optimal scheme under the same state. With this definition, the effectiveness of different schemes can be determined. Suppose that after $k$ times of information collection, the probabilities of all states being true are $\mu(T_1|h_k)$, $\mu(T_2|h_k)$, ..., and $\mu(T_j|h_k)$. Under such a circumstance, the decision loss function $L(h_k, P_i)$ using the scheme $P_i$ can be expressed as:

$$L(h_k, P_i) = \sum_{j=1}^{J} \mu(T_j|h_k)B_{ji} \tag{7}$$

By calculating the decision loss functions of different schemes under a certain state, the decision-maker can identify the scheme $P_i^m$ with the smallest decision loss. With the scheme with the minimum decision loss at a certain stage identified, the decision-maker can compare the cost of information collection and the calculated result of the anticipated decision loss function during the next stage to determine whether it is necessary to continue to collect information at the next stage. The anticipated decision loss function, namely, the next anticipated decision loss function after $k$ times of information collection, is denoted as $L_e(h_k, P_i)$, and this anticipated decision loss function can also be calculated using Formula (7).

With the continuous collection of information, the calculated result of the current decision loss function is compared with the calculated result of the anticipated decision loss function at the next stage and the cost of continuous information collection. Once the following condition is satisfied, the information collection should be stopped:

$$L(h_k, P_i) \leq L_e(h_k, P_i) + C_{k+1} \tag{8}$$

Through the calculation of information collection costs and decision loss functions, it can be decided whether to collect information continuously or directly adopt the current solution scheme that is matched. Therefore, such a dilemma that decisions made too early will result in decision errors and decisions made too late will bring about the loss of decision opportunity can be solved. Thus, spatiotemporal solution schemes can be optimized.

## 5. Methods of On-Site Emergency Decision-Making in Major Accidents

During the handling processes of major accidents, information is incomplete and uncertain. In conventional sequential decision-making methods, expert judgment is heavily relied on. Therefore, once experts make the wrong judgment, there could be poor accident handling effects or even deteriorated on-site situations. In view of this, this paper proposes an on-site emergency decision-making method for major accidents with the situation prediction model based on the D-S evidence theory combined with the method of sequential decision-making. With this method, evolution directions of major accidents with incomplete information can be predicted, and possible accident states at the next stage can be judged. Therefore, the solution schemes can be better determined.

### 5.1. Analysis of On-Site Game Processes of Major Accidents

Bayesian theory is generally used in probability calculations in the processes of sequential games [21]. The general idea applied in sequential decision-making of major accidents using the Bayesian theory can be briefly described as follows: At the early stages of major accidents, information on the accident scenes is complicated, with no effective information available. Therefore, decision-makers must judge the accident states at the current stage based on historical experiences, thus obtaining prior probabilities of these

accident states. With the development and evolution of accidents, improved accident information will be gradually collected, and decision-makers will modify the conditional accident probabilities and obtain the posterior probabilities of accident states. With major accident and emergency decision-makers taken as the two sides of the game, the game process of emergency decision-making can be described as follows: Let $t_0$ be the occurrence moment of a major accident, and $t_1$ and $t_2$ be the first and the second decision-making moments of the emergency decision-makers, respectively.

(1) At the $t_0$ moment, the initial state of the major accident is settled with a "natural" selection.

(2) During the period of $t_0$–$t_1$, the condition of the major accident further deteriorates, with property losses and possible casualties caused.

(3) At the $t_1$ moment, based on the preliminary information obtained, the emergency decision-maker determines the prior probability of the major accident state at this moment. At the same time, an emergency expert group is convened for the major accident. Based on the information collected, the expert group judges and determines the conditional probability of each key accident scenario so as to calculate the posterior probabilities of accident states. Following the principle of maximum expected utility, the expert group chooses the most suitable emergency scheme for immediate implementation.

(4) During the period of $t_1$–$t_2$, under the joint effect of the disposal scheme and evolution rule, the major accident will evolve into a new state.

(5) Similarly, at the $t_2$ moment, the emergency decision-maker evaluates the accident state at the current stage. Through the collection and analysis of accident information, the decision-maker uses the posterior probability of the accident state at the previous stage as the prior probability of the accident state at the current stage to calculate the posterior probability of the current accident state, and then selects an appropriate rescue scheme for implementation.

(6) Steps (4) and (5) are repeated until the major accident is completely under control and its impacts are significantly alleviated.

The on-site sequential game flow of a major accident under a conventional mode is shown in Figure 4.

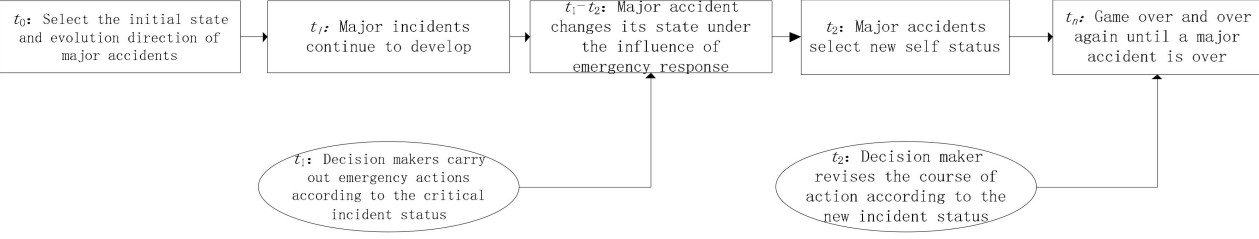

**Figure 4.** Game flow of sequential decision-making of a major accident.

*5.2. Emergency Decision-Making Processes of Major Accidents*

In order to effectively avoid the subjective biases of expert groups and provide more accurate state information for decision-makers, a situation prediction model based on the D-S evidence theory is introduced in this paper. The aim is to analyze the next-stage key accident scenarios of major accidents more accurately through scenario symptoms to predict the accident states at the next stage. Therefore, through the calculations of decision-making profit and loss functions, different time and space solution schemes can be better determined. The basic idea is as follows:

(1)   At the initial moment, the decision-maker is informed of the occurrence of a major accident, and at this very moment, the decision-maker needs to make a judgment with almost no information. Additionally, the decision-maker needs to extract useful information from a brief description of the accident state, compare historical cases, assume the on-site accident situation, and formulate a solution scheme. Assume that there occurs a chemical accident. Firstly, according to the accident evolution situation,

it is assumed that the state of the accident scene could be one of four states, namely $T_1$, $T_2$, $T_3$, and $T_4$, which are briefly described as follows:

$T_1$: A major danger is about to happen on the scene,
$T_2$: Temporarily, there is no major danger risk at the scene of the accident,
$T_3$: The accident state tends to stabilize,
$T_4$: The accident gradually vanishes.

Four disposal schemes corresponding to these four possible accident states mentioned above are designated. Denoted as $P_1$, $P_2$, $P_3$, and $P_4$, these four schemes are briefly described as follows:

$P_1$: Continuously observe the state of the disaster and prevent casualties.
$P_2$: Take all necessary measures to control the accident in a stable state.
$P_3$: Launch an all-out attack to make the accident gradually vanish.
$P_4$: Go to the accident site to deal with the accident.

(2) With data obtained from the accident database or historical cases, a profit and loss matrix and an accident loss probability distribution table should be created. Assume that the cost of information collection at each stage is one unit ($C_k = 1, k = 1, 2, \ldots, N$), and the sample spaces represent the degrees of losses caused by accident, namely, $I_1$ (particularly serious), $I_2$ (serious), $I_3$ (relatively serious), and $I_4$ (general serious). Based on the field situation, the accident states and the benefits and losses caused by the corresponding solution schemes are listed in Table 3.

**Table 3.** Profit and loss matrix.

| | | Accident State | | | | | | | |
|---|---|---|---|---|---|---|---|---|---|
| | | $T_1$ | | $T_2$ | | $T_3$ | | $T_4$ | |
| | | **Benefit** | **Loss** | **Benefit** | **Loss** | **Benefit** | **Loss** | **Benefit** | **Loss** |
| | $P_1$ | 100 | 0 | −50 | 150 | −100 | 200 | −150 | 250 |
| Solution | $P_2$ | −200 | 300 | 100 | 0 | −50 | 150 | −100 | 200 |
| scheme | $P_3$ | −300 | 400 | −200 | 300 | 100 | 0 | −50 | 150 |
| | $P_4$ | −400 | 500 | −300 | 400 | −200 | 300 | 100 | 0 |

The most effective disposal measure corresponding to the specific accident state needs to be used in the field of a major accident. For example, under an accident state of $T_1$, if a firefighting attack instead of emergency avoidance is carried out, huge losses could be caused. Table 4 shows the loss probabilities of different schemes corresponding to different accident states developed based on historical data [22–24].

**Table 4.** Probability distribution of accident losses.

| | $T_1$ | $T_2$ | $T_3$ | $T_4$ |
|---|---|---|---|---|
| $I_1$ | 0.6 | 0.2 | 0.15 | 0.05 |
| $I_2$ | 0.2 | 0.6 | 0.2 | 0.15 |
| $I_3$ | 0.15 | 0.2 | 0.6 | 0.2 |
| $I_4$ | 0.05 | 0.15 | 0.2 | 0.6 |

(3) A first round of decision-making is initiated, and decision-making losses of those four schemes of $P_1$, $P_2$, $P_3$, and $P_4$ corresponding to four accident states of $T_1$, $T_2$, $T_3$, and $T_4$ are calculated, respectively. By comparing these decision-making losses with calculated results of decision-making loss functions, the decision-maker needs to decide when and which solution scheme should be adopted and simultaneously start the first-round information collection. During the information collection process, with sufficient key accident scenarios and accident scenario symptoms obtained, the decision-maker needs to judge the current accident state and use the situation

prediction model based on the D-S evidence theory to predict the next-stage accident state. Based on the situation prediction results, the decision-maker should calculate the anticipated decision loss at the next stage and compare the calculated result of the decision loss function at the current stage with the calculated result of the anticipated decision loss function to make the emergency decision for the current stage.

(4)    Step (3) should be repeated until the accident completely vanishes.

(5)    Application example: The emergency decision-making of key accident scenarios of the "P.X. project explosion accident in Zhangzhou, China, on 6 April" was taken as an example.

At 18:54 on 6 April 2015, due to welding defects, an adsorption and separation device broke in the Gu Lei Petrochemical Tenglong Aromatics Co., Ltd. in Zhangzhou City, China, resulting in a fire in the adsorption and separation device and three storage tanks in the intermediate tank farm. After the fire broke out, all department levels quickly started the contingency plan and coordinated the accident disposal on the scene. At 16:40 on 7 April, the open flame was extinguished for the first time. After that time, fires were rekindled several times. On 15 April, after firefighting techniques were applied and residual liquids were transferred and disposed of, the on-site danger was eliminated, and the firefighting and rescue work was accomplished.

After receiving the alarm, the Gu Lei Brigade of Zhangzhou detachment quickly responded and dispatched ten vehicles and sixty firefighters to the scene and reported the disaster situation to the detachment, the headquarters, the provincial public security department, and the command center of the department and bureau, step-by-step. At the same time, the command center of the headquarters also informed the duty room of the provincial government about the situation.

After the rescue force received the alarm and before it arrived at the accident scene: When receiving the alarm, normally decision-makers do not have any information. Based on historical cases, it can be assumed that the accident at that moment could be at one of four stages, namely $T_1$, $T_2$, $T_3$, and $T_4$, which are described as follows:

$T_1$: The accident is at a stage of violent growth, and there could occur a major danger. Thus, emergency danger avoidance measures should be employed.

$T_2$: The accident is in a serious state, but there is no risk of major danger. Therefore, relevant measures should be taken to control the accident in a stable state as much as possible.

$T_3$: The accident is in a relatively stable state, and certain measures can be taken to reduce the impacts of the accident.

$T_4$: The accident state is less threatening. Therefore, the rescue force can go deep into the accident site for disposal.

At this moment, the decision-maker cannot determine the actual state of the accident. Therefore, it is assumed that each accident state has an equal probability. That is, $\mu(T_1|\varphi) = 0.25$, $\mu(T_2|\varphi) = 0.25$, $\mu(T_3|\varphi) = 0.25$, and $\mu(T_4|\varphi) = 0.25$.

Four disposal schemes corresponding to the four possible accident states are designated. These four schemes, namely $P_1$, $P_2$, $P_3$, and $P_4$, are briefly described as follows:

$P_1$: After arriving at the scene, decision-makers judge the safe distance, continuously observe the disaster situation by means of remote investigation, and prepare to deal with any major danger that could occur.

$P_2$: After arriving at the scene, decision-makers control the accident in a stable state through measures of setting up positions and extinguishing fires with cooling.

$P_3$: After arriving at the scene, decision-makers put all forces into the firefighting and rescue activity, thus making the accident gradually vanish.

$P_4$: After analyzing the on-site situation, decision-makers send the rescue force deep into the site to handle the accident with techniques of danger elimination and methods of fire extinguishing from the top.

Therefore, the decision-making loss function of each scheme can be calculated as follows:

Scheme $P_1$: $L(h_0, P_1) = B_{11}\mu(T_1|\varphi) + B_{12}\mu(T_1|\varphi) + B_{13}\mu(T_1|\varphi) + B_{14}\mu(T_1|\varphi) = 0 \times 0.25 + 150 \times 0.25 + 200 \times 0.25 + 250 \times 0.25 = 150$

Scheme $P_2$: $L(h_0, P_2) = 162$
Scheme $P_3$: $L(h_0, P_3) = 212.5$
Scheme $P_4$: $L(h_0, P_4) = 300$

It can be seen that scheme $P_1$ is the optimal scheme for the moment in question. That is, after arriving at the scene, decision-makers should judge the safe distance, continuously observe the disaster situation by means of remote investigation, and prepare to deal with any major danger that could occur. Suppose that after arriving at the accident scene, the rescue force starts the first round of information collection ($h_1$). Additionally, suppose that the loss incurred in each accident state has an equal probability. That is, $\mu(I_1) = \mu(I_2) = \mu(I_3) = \mu(I_4) = 0.25$. Then, the value of the expected posterior probability of each accident state can be calculated as follows: $\mu(T_1|h_1) = 0.261$, $\mu(T_2|h_1) = 0.241$, $\mu(T_3|h_1) = 0.240$, and $\mu(T_4|h_1) = 0.258$. Therefore, the expected decision-making loss functions of those four solution schemes applied at that moment can be calculated as follows:

Scheme $P_1$: $L_e(h_1, P_1) = B_{11}\mu_e(T_1|h_1) + B_{12}\mu(T_2|h_1) + B_{13}\mu(T_3|h_1) + B_{14}\mu(T_4|h_1) = 148.65$
Scheme $P_2$: $L_e(h_1, P_2) = 165.9$
Scheme $P_3$: $L_e(h_1, P_3) = 215.4$
Scheme $P_4$: $L_e(h_1, P_4) = 298.9$
Additionally, $L_e(h_0, P_i) = P_i^m = 148.65$.

A comparison of these calculated values of loss functions shows that the loss incurred with the direct implementation of scheme $P_1$ by emergency decision-makers without information collection is higher than that incurred with the implementation of scheme $P_1$ by decision-makers after one round of information collection. That is, with a lack of information, decision-makers could make less precise decisions. Therefore, decision-makers should continue to collect information.

The first arrival of the disposal force at the accident scene: At 19:03 on 6 April, based on the on-site investigation, the precinct brigade found that the adsorption and separation device of aromatic hydrocarbon had exploded, and the No. 607, No. 608, and No. 610 tanks in the intermediate tank farm on the west side of the device cracked and burned violently. That is, the first-round information collection obtained a result of $I_2$ (the information collected indicated an accident state of $T_2$ at the current stage). Therefore, the values of the actual posterior probabilities at this moment can be calculated as follows: $\mu(T_1|I_2) = 0.18$, $\mu(T_2|I_2) = 0.54$, $\mu(T_3|I_2) = 0.14$, and $\mu(T_4|I_2) = 0.14$. Thus, the calculated values of decision-making loss functions of those four solution schemes applied at this moment are as follows: 144 (Scheme $P_1$), 103 (Scheme $P_2$), 255 (Scheme $P_3$), and 348 (Scheme $P_4$). Suppose that a second-round information collection ($h_1$) was performed, with the following values of expected posterior probabilities obtained: $\mu(T_2|h_2) = 0.236$, $\mu(T_2|h_2) = 0.272$, $\mu(T_3|h_2) = 0.254$, and $\mu(T_4|h_2) = 0.238$. Under such a circumstance, the expected decision-making loss function obtains a minimum value of 175.2. This indicates that it is not necessary for decision-makers to collect information continuously at this moment. Instead, decision-makers should employ the scheme $P_2$ directly. That is, after arriving at the accident scene, decision-makers should control the accident in a stable state through such measures as setting up positions and extinguishing fires with cooling.

Stalemate stage of accident disposal: With the continuous evolution of the on-site situation of the oil tank fire and the development of firefighting and rescue operations, as well as the deepening of investigation work, firefighters and rescuers can obtain multiple accident symptoms to form an accident symptom space. Therefore, the situation prediction model based on the D-S evidence theory can be used by decision-makers to predict the evolution patterns of major accidents and guide their decision-making in the next step. The time series of key accident scenario symptoms in the field are listed in Table 5.

**Table 5.** Time series of key accident scenario symptoms.

| Time | Key Accident Scenario Symptoms |
|---|---|
| 19:03, 6 April | The adsorption and separation device of aromatic hydrocarbon exploded, and the No. 607, No. 608, and No. 610 tanks cracked and burned fiercely, with a high flame temperature and strong radiation heat. The explosive combustion led to the leakage fires of some pipelines. |
| 23:40, 6 April | A fire broke out in the main power station of the coal-fired power generation facility in the south of the factory. |
| 02:50, 7 April | The rubber sealing ring on top of the No. 102 external floating roof tank caught fire. |
| 09:50, 7 April | The fire of the No. 607 tank was extinguished, and at about 10:25, the fire of the No. 608 tank was put out. |
| 11:30, 7 April | The No. 608 oil tank was rekindled. |
| 17:05, 7 April | The open flame of the No. 610 burning tank was completely extinguished. By 19:00, the wall temperatures of the No. 607, No. 608, and No. 610 tanks were 21 °C, 24 °C, and 24 °C, respectively. The top temperature of the No. 610 tank was 60 °C, and the temperature of each tank dropped rapidly. |
| 19:40, 7 April | The oil surface of the No. 610 tank was rekindled. |

Based on the time series of on-site key accident scenario symptoms, key accident scenarios were analyzed as follows:

At 19:03 on 6 April, the adsorption and separation device exploded and burned, and the No. 607, No. 608, and No. 610 oil tanks burned violently, with a high flame temperature and strong radiation heat. The explosive combustion resulted in leakage fires of some pipelines, and the wind on the accident scene turned from a northerly wind to an easterly wind, with a speed of category two and a gust of category four. Based on this field situation, the evidence of a "pipeline leakage fire" has been confirmed. Therefore, its confidence level was assigned with a value of 1. However, with the dangerous on-site situation, as for the evidence of "fierce combustion, high flame temperature, and strong radiation heat", it was not possible for decision-makers to go deep into the tank farm to obtain accurate data. Therefore, its confidence level was assigned with a value of 0.4. From Table 2, it can be seen that key accident scenarios, which correspond to the evidence of "pipeline leakage fire" and "fierce combustion, high flame temperature, and strong radiation heat" and could occur on the scene of the accident, were as follows: "flowing fire" ($S_1$), "pipeline spewing fire" ($S_2$), "explosive combustion of the adjacent tank" ($S_4$), and "tank secondary explosion" ($S_5$). Based on the field situation, the key accident scenario of "flowing fire" ($S_1$) was less likely to occur at the accident scene because the adsorption and separation device had burned steadily. Thus, the reverse probability of this scenario was assigned with a value of 0.2. Similarly, the reverse probabilities of key accident scenarios of "pipeline spewing fire", "explosive combustion of the adjacent tank", and "tank secondary explosion" were assigned with values of 0.1, 0.1, and 0.2, respectively.

With the synthesis calculation in Formulas (1)–(5) based on the D-S evidence theory, the following calculated values of probability distribution functions of all key accident scenarios mentioned above were obtained: $S_1$ (0.702, 0.80), $S_2$ (0.652, 0.90), $S_4$ (0.1362, 0.980), and $S_5$ (0.0795, 0.980).

Through probability distribution functions, the probabilities of four key accident scenarios that may occur on the scene of the accident were obtained. The calculated value of the probability distribution function of scenario $S_1$, (0.702, 0.80), indicates that the practical significance of this function lies in the fact that the decision-maker believes $S_1$ will occur with the lowest certainty level of 0.702 and the highest certainty level of 0.80. According to the situation prediction model constructed in this paper, decision-making principles themselves will also affect the results of decision-making. That is, in terms of judged thresholds of probability functions, active decision-makers have a high threshold, while

conservative decision-makers have a low threshold. Due to unclear accident situations and an insufficient rescue force, it is assumed that decision-makers at accident scenes are relatively conservative. Therefore, in this paper, the threshold was set at 0.7. Figure 5 shows the constructed coordinate axis of threshold determination.

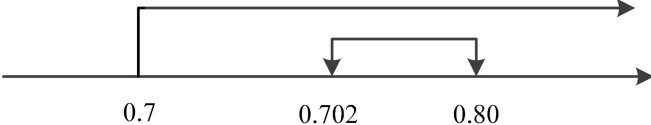

**Figure 5.** Coordinate axis of threshold determination.

From the coordinate axis of threshold determination shown in Figure 5, it can be seen that the values of the probability distribution function of $S_1$ as a whole were located on the right side of the threshold. Therefore, it is very likely that the $S_1$ scenario will occur. Additionally, the calculated probability of this scenario was (0.1), indicating that its occurrence probability is 1. Similarly, analysis results of $S_2$, $S_4$, and $S_5$ were obtained, which are listed in Table 6.

**Table 6.** Information table of the first-round accident situation prediction (at 19:03 on 6 April).

| Key Accident Symptom | Key Accident Scenario | Probability Distribution Function | Calculated Probability |
|---|---|---|---|
| Pipeline leakage fire | Flowing fire, $S_1$ | (0.702, 0.80) | (0.1) |
| | Pipeline spewing fire, $S_2$ | (0.652, 0.90) | (0.19, 0.81) |
| Fierce combustion, high flame temperature, and strong radiation heat | Explosive combustion of adjacent tank, $S_4$ | (0.1362, 0.980) | (0.67, 0.33) |
| | Tank secondary explosion, $S_5$ | (0.0795, 0.980) | (0.1) |

Table 6 shows that, according to the on-site accident situation, it can be judged that scenarios $S_1$, $S_2$, and $S_5$ were more likely to occur, and scenario $S_4$ was less likely to occur. Therefore, those four initial disposal schemes were adjusted accordingly, with the following calculation results of the decision-making loss and expected loss of each scheme:

$$L(h_3, P_3) = 143.2 < L_e(h_1, P_3) = 184.5$$

Therefore, it can be determined that information collection is not necessary under such a circumstance. Instead, an immediate decision should be made, and multiple measures should be taken to launch an all-around attack to make the major accident gradually vanish. The actual accident situation is that, at 7:00 on 7 April, there was a sufficient rescue force at the accident scene. In order to eliminate the threat of those three burning tanks to the No. 609 tank and prevent these tanks from being rekindled, the on-site command post decided to launch a firefighting attack. Specifically, the cooling protection of the No. 609 tank was strengthened, and the fires of the No. 607, No. 608, and No. 610 burning tanks were put out, in turn, from the headwind direction to the downwind direction.

Vanishing stage of the accident: At 17:05 on 7 April, the open flame of the No. 610 burning tank was completely extinguished. Before that, the fires of the No. 607 and No. 608 tanks were also put out. During this period, the No. 608 tank was rekindled. By 19:00, the wall temperatures of the No. 607, No. 608, and No. 610 tanks were 21 °C, 24 °C, and 24 °C, respectively. The top temperature of the No. 610 tank was 60 °C, and the temperature of each tank dropped rapidly. The meteorological conditions at that moment are as follows: easterly wind, with an average wind speed of category four and a gust of category seven. Table 7 shows the updated prediction details of the accident situation with the No. 610 tank taken as the research object.

**Table 7.** Information table of the second-round accident situation prediction (at 17:05 on 7 April).

| Key Accident Symptom | Key Accident Scenario | Probability Distribution Function | Threshold Analysis Result |
|---|---|---|---|
| Not favorable wind direction for the on-site firefighting and rescue activity | Tank secondary explosion, $S_5$ | (0.415, 0.682) | (1.0) |
| | Tank rekindling, $S_6$ | (0.626, 0.95) | (0.23, 0.77) |
| The oil flame point is relatively low | Spreading toxic gas, poisoning the environment, $S_7$ | (0.619, 0.728) | (0.47, 0.53) |

From Table 7, it can be seen that, basically, the tanks would not explode. However, it was more likely that the tanks would be rekindled. Since there was no toxic gas at the scene, it was not necessary to consider the problem of toxic gas spreading. Based on the second-round situation prediction results, the recommended emergency solution was that the No. 610 tank should be continuously cooled to prevent it from being rekindled. The real situation was that, at 19:40 on 7 April, the foam coating on the oil surface of the No. 610 tank was destroyed by strong wind and rain, and the high-temperature oil in the tank was exposed to the air and rekindled. Then, the command post immediately organized the on-site force to enhance cooling of the tank, which is basically consistent with the proposed solution described above.

Follow-up disposal: By 12:00 on 8 April, the on-site command post used an unmanned aerial vehicle to survey the situation of the intermediate tank farm, with the following information obtained: The materials in the No. 609 tank had been basically burned out, and there were still residual liquids in the No. 607, No. 608, and No. 610 tanks. Based on the information collected, it can be known that the on-site situation at that moment was $T_4$, and with the decision-making function loss compared with the expected decision-making loss, it can be concluded that the scheme $P_4$ should be directly implemented at that moment. The real situation was that, based on the field situation, the command post decided to continuously monitor the accident site and organize forces to eliminate the danger in a technical way. By 15 April, residual liquids in the tanks were basically drained, and the on-site danger was eliminated.

## 6. Conclusions

Based on D-S evidence theory and game theory, we investigated the sequential decision-making method of major accidents with incomplete information, with main conclusions drawn as follows:

(1) With databases of historical cases analyzed and an oil tank fire accident taken as an example, this paper has established the logical relationship between key accident scenarios and accident scenario symptoms, with the basic probability distribution of an evidence set of oil tank fire accidents obtained through the logistic regression analysis. Meanwhile, this paper showed that the relationship between key accident scenarios and key accident scenario symptoms of major accidents can be analyzed in a quantitative rather than a qualitative way, which can improve the accuracy of emergency decision-making information.

(2) The situation prediction model based on the D-S evidence theory can fuse and process the accident information under different time series of major accident scenes and correct the subjective prediction biases of experts with incomplete information, thus resulting in more scientific and reliable emergency decision-making. Therefore, accident situations can be predicted dynamically.

(3) With the situation prediction model based on the D-S evidence theory applied in the sequential decision-making of major accidents, the problem of difficult scheme optimization at different time stages caused by the addition of time costs at the scenes

of major accidents, as well as the contradiction between decision-making loss costs and information collection costs, can be resolved.

(4) The decision-making model established in this paper solely focused on the analysis of petrochemical fires, while further research is needed to establish case databases for other types of disasters and analyze their decision-making processes. Additionally, the traditional methods used for information collection and processing are inefficient and inadequate to support the decision-making model proposed in this paper. However, future advancements in big-data-related technologies can potentially provide support for the decision-making model.

**Author Contributions:** Conceptualization, D.X. and J.X.; methodology, Y.Z. and C.C.; data curation, C.Z. and C.C.; investigation, C.C. and C.Z.; writing—original draft preparation, C.Z.; writing—review and editing, C.C.; supervision, D.X.; funding acquisition, D.X. All authors have read and agreed to the published version of the manuscript.

**Funding:** This research was funded by [National Natural Science Foundation of China] grant number [No. 52174224].

**Institutional Review Board Statement:** Not applicable.

**Informed Consent Statement:** Informed consent was obtained from all subjects involved in the study.

**Data Availability Statement:** Data are contained within the article.

**Acknowledgments:** We are grateful to the editor and anonymous reviewers for their valuable comments, which improved this paper significantly.

**Conflicts of Interest:** The authors declare no conflicts of interest.

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
