# Peer review of "Research on Sequential Decision-Making of Major Accidents with Incomplete Information"

_fire, doi:10.3390/fire7020049_

Round 1
Reviewer 1 Report
Comments and Suggestions for Authors
Review report
Manuscript ID: fire-2844169
Title: Research on Sequential Decision-Making of Major Accidents with Incomplete Information
Authors
Dengyou Xia , Changlin Chen * , Ce Zheng , Jing Xin , Yi Zhu
The paper proposes a method of sequential decision-making using the relevant knowledge of D-S evidence theory and game theory. Based on the D-S evidence theory, the proposed method uses approximate reasoning on the incomplete and insufficient information obtained at the scenes of major accidents.
Comments and suggestions.
· Write the limitations of the proposed method in the conclusion section.
· References are not uniformly written for journal articles. Check them and fix them. For examples, check [2], [4], [5], [6], [7], [19], [22].
Comments on the Quality of English Language
Review report
Manuscript ID: fire-2844169
Title: Research on Sequential Decision-Making of Major Accidents with Incomplete Information
Authors
Dengyou Xia , Changlin Chen * , Ce Zheng , Jing Xin , Yi Zhu
The paper proposes a method of sequential decision-making using the relevant knowledge of D-S evidence theory and game theory. Based on the D-S evidence theory, the proposed method uses approximate reasoning on the incomplete and insufficient information obtained at the scenes of major accidents.
Comments and suggestions.
· Write the limitations of the proposed method in the conclusion section.
· References are not uniformly written for journal articles. Check them and fix them. For examples, check [2], [4], [5], [6], [7], [19], [22].
Author Response
Thank you very much for taking the time to review this manuscript.
1. A description of limitations has been added to the conclusion.
2. The format of the references has been adjusted

Reviewer 2 Report
Comments and Suggestions for Authors
In order to solve the problem of emergency decision-making with
incomplete information and deal with the accident information in different
time series at the scenes of major accidents, this paper has proposed a
method of sequential decision-making by utilizing the relevant knowledge of
D-S evidence theory and game theory. Based on the D-S evidence theory, this method employs approximate reasoning on the incomplete and insufficient information obtained at the scenes of major accidents, thus realizing the situation prediction of key scenes of these accidents. Also, this method uses the game theory to solve the contradiction between decision-making loss cost and information collection cost, thus optimizing the decision-making schemes at different time stages of major accidents. Firstly, this paper has taken an oil-tank fire accident as an example and sorted out historical cases and expert experiences to establish a logical relationship between key accident scenes and accident scene symptoms in the accident. Meanwhile, it has applied the Logistic regression analysis method to obtain the basic probability distribution of each key accident scene in the oil-tank fire, and on this basis, it has constructed an evidence set of the fire. Secondly, based on the D-S evidence theory, this paper has effectively quantified the knowledge uncertainty and evidence uncertainty, with the incomplete and insufficient information taken as an evidence system of the development of key accident scenes to construct a situation prediction model of these accident scenes. Thirdly, based on the game theory, this paper has viewed emergency decision-makers and major accidents as two sides of the game to compare and analyze accident states at different time points and solve the contradiction between loss costs of decision-making and information collection costs.
Therefore, this paper has provided a solution for the optimization of accident schemes at different time stages, thus realizing the sequential decision-making at the scenes of major accidents. Furthermore, this paper has combined the situation prediction model with sequential decision-making, with the basic steps described below: (1) Drawing up an initial action plan in the case of extreme lack of information; (2) Adjusting the initial action plan
with partial information collected to construct a framework of accident
identification; (3) Starting to address the accident and deal with the
continuously added evidence information with the evolution of the accident;
(4) Calculating the confidence levels of key accident scenarios after
evaluating different evidence and then predicting the accident state in the
next stage; (5) Calculating the profit-loss ratio between current
decision-making scheme and decision-making scheme of the next stage; (6)
Repeating Steps (4) and (5) until the accident completely vanishes. Finally,
this paper has verified the feasibility of the proposed method with the
explosion accident of the Zhangzhou P.X. project in Fujian on April 6th used
as an example.
The article is relevant, especially in our difficult time.
But the article has a number of disadvantages:
- The title of the article is incorrect. I recommend the following title of the article "Game and System Methods of Researching the Decision-Making Procedure in the Elimination of Major Accidents with Incomplete Information on the Consequences and Estimates of Material and Human Losses"
- Тhe cognitive features of operational managers involved in the elimination of accidents are not taken into account;
- Іt is not specified what types of accidents (man-made, environmental, etc.) this problem is focused on;
- Тhe impact of accidents on the environment and their consequences is not specified;
- With different types of accidents, there are a number of risks in their elimination. No risk is given.
- The decision-making process in possible and real situations is not separated.
- According to the purpose of the study, the article is not structured and the goal is not related to the structure of the decision-making process (goal, objectives).
- The players in the emergency system and the factors that led to the destructive factors have not been identified.
- The system of logical inferences does not clearly connect - the object, situation, threats, risks, losses - with the target task of the study.
- Diagram 1 does not link the influencing factors and the logic of the game, rather than the risk assessment.
- Figure 2 does not explain the result of logical inference in terms of ranking risks in specific situations.
- The game course of sequential decision-making about a major accident for what purposes – assessment of the level of risk for the forecast, assessment of losses, assessment of the prerequisites to prevent an emergency, forecast of possible schemes to counteract the occurrence of accidents.
- The article should be structured according to the main purpose of the study.
Comments on the Quality of English Language
the article is relevant especially
Author Response
Thank you very much for taking the time to review this manuscript.
Through the discussion of our research group, I will give you some feedback on our understanding of this article.
1. Your suggestion is very good, but the core of this paper is the emergency decision-making method, which does not consider the consequences of accidents and losses.
2. This is the shortcoming of this paper, and we will strengthen the research in this area in the future work.
3. The decision method proposed in this paper is applicable to all major accidents. In this paper, only major accidents in the petrochemical industry are taken as an example.
4-5. Thank you for your comments. We will strengthen the research in this area in the following work.
6. An important part of the decision-making method proposed in this paper is to compare the profit-loss ratio between the current decision and the expected decision.
8. Thank you for your comments. We will strengthen the research in this area in the following work.
9. We have tried our best to unify and quantify these contents in the paper, but the research is not in-depth enough, so it may not achieve the effect you expect.
10. The statistical results of the data are shown in Table 1.
11. The specific meaning of Figure 2 has been explained in the preceding paragraphs.
7、12、13.Thank you for your valuable comments. The structure of this paper is developed through the actual emergency treatment process, and it may indeed be difficult to understand the structure.
Reviewer 3 Report
Comments and Suggestions for Authors
The manuscript with the title: “Research on Sequential Decision-Making of Major Accidents with Incomplete Information" is designed very appropriately. And the different parts of the article are well compiled. The conclusion section as well as the "discussion" section are well prepared. Therefore, there are only two minor comments to complete the presented article:
1- The abstract of the article should be a little summarized and some unnecessary sentences should be removed.
2- The following references are also necessary to be added to the background section of the research:
2-1- A mathematical model and application for fire risk management in commercial complexes in South Africa, Results in Engineering, Volume 7, 2020. https://doi.org/10.1016/j.rineng.2020.100145
2-2-Risk and incident analysis on key safety performance indicators and anomalies feedback in south pars gas complex. Results in Engineering. 9, 100210. https://doi.org/10.1016/j.rineng.2021.100210
Comments on the Quality of English LanguageThere is no conflict of interest.
Author Response
Thank you very much for taking the time to review this manuscript.
1.The abstract of the article has been revised according to your comments
2.References have been added

Round 2
Reviewer 1 Report
Comments and Suggestions for Authors
Journal: Fire (ISSN 2571-6255)
Manuscript ID: fire-2844169
Title: “Research on Sequential Decision-Making of Major Accidents with Incomplete Information”
Authors: Dengyou Xia, Changlin Chen, Ce Zheng, Jing Xin, Yi Zhu
The authors revised the manuscript. However, further revision is required.
· References are not uniformly written. Please check all the references and fix the issues.
o For example, replace “Research on assistant decision-making method for railway emergency management” with “research on assistant decision-making method for railway emergency management” in reference [2].
o Similar changes are required for the references [4, 5, 6, 7, 14, 15, 16,22, 24]
Comments on the Quality of English Language
Journal: Fire (ISSN 2571-6255)
Manuscript ID: fire-2844169
Title: “Research on Sequential Decision-Making of Major Accidents with Incomplete Information”
Authors: Dengyou Xia, Changlin Chen, Ce Zheng, Jing Xin, Yi Zhu
The authors revised the manuscript. However, further revision is required.
· References are not uniformly written. Please check all the references and fix the issues.
o For example, replace “Research on assistant decision-making method for railway emergency management” with “research on assistant decision-making method for railway emergency management” in reference [2].
o Similar changes are required for the references [4, 5, 6, 7, 14, 15, 16,22, 24]